# Asymmetric Event-Guided Video Super-Resolution

Zeyu Xiao*
MoE Key Laboratory of
Brain-inspired Intelligent Perception
and Cognition, University of Science
and Technology of China
Hefei, China
zeyuxiao@mail.ustc.edu.cn

Dachun Kai*
MoE Key Laboratory of
Brain-inspired Intelligent Perception
and Cognition, University of Science
and Technology of China
Hefei, China
dachunkai@mail.ustc.edu.cn

Yueyi Zhang†
MoE Key Laboratory of
Brain-inspired Intelligent Perception
and Cognition, University of Science
and Technology of China
Hefei, China
zhyuey@ustc.edu.cn

Xiaoyan Sun
MoE Key Laboratory of
Brain-inspired Intelligent Perception
and Cognition, University of Science
and Technology of China
Hefei, China
sunxiaoyan@ustc.edu.cn

Zhiwei Xiong
MoE Key Laboratory of
Brain-inspired Intelligent Perception
and Cognition, University of Science
and Technology of China
Hefei, China
zwxiong@ustc.edu.cn

## Abstract

Event cameras are novel bio-inspired cameras that record asynchronous events with high temporal resolution and dynamic range. Leveraging the auxiliary temporal information recorded by event cameras holds great promise for the task of video super-resolution (VSR). However, existing event-guided VSR methods assume that the event and RGB cameras are strictly calibrated (*e.g.*, pixel-level sensor designs in DAVIS 240/346). This assumption proves limiting in emerging high-resolution devices, such as dual-lens smartphones and unmanned aerial vehicles, where such precise calibration is typically unavailable. To unlock more event-guided application scenarios, we perform the task of asymmetric event-guided VSR for the first time, and we propose an Asymmetric Event-guided VSR Network (AsEVSRN) for this new task. AsEVSRN incorporates two specialized designs for leveraging the asymmetric event stream in VSR. Firstly, the content hallucination module dynamically enhances event and RGB information by exploiting their complementary nature, thereby adaptively boosting representational capacity. Secondly, the event-enhanced bidirectional recurrent cells align and propagate temporal features fused with features from content-hallucinated frames. Within the bidirectional recurrent cells, event-enhanced flow is employed to simultaneously utilize and fuse temporal information at both the feature and pixel levels. Comprehensive experimental results affirm that our method consistently generates superior quantitative and qualitative results. The code is publicly available at: https://github.com/zeyuxiao1997/AsEVSRN.

---
*Zeyu and Dachun contribute equally to this work.
†Corresponding author.

## CCS Concepts

• **Computing methodologies → Reconstruction**.

## Keywords

Video Super-Resolution, Event Camera, Stereo Images

**ACM Reference Format:**
Zeyu Xiao, Dachun Kai, Yueyi Zhang, Xiaoyan Sun, and Zhiwei Xiong. 2024. Asymmetric Event-Guided Video Super-Resolution. In *Proceedings of the 32nd ACM International Conference on Multimedia (MM '24), October 28-November 1, 2024, Melbourne, VIC, Australia.* ACM, New York, NY, USA, 10 pages. https://doi.org/10.1145/3664647.3681357

## 1 Introduction

High-resolution (HR) videos are attracting increasing attention in both academia and industry, and have been already widely used in modern society, especially for the multimedia field. Video super-resolution (VSR) stands as a foundational task within the domains of computer vision to generate HR videos. The primary goal of VSR is to enhance visual quality by reconstructing an HR video from a low-resolution (LR) observation. VSR has garnered substantial attention and popularity due to its diverse applications, encompassing areas such as video surveillance [1, 77], high-definition television [12], and satellite imagery [8, 38, 62, 64]. In contrast to single-image super-resolution, which primarily addresses spatial dimensions, VSR uniquely exploits both spatial and temporal dependencies. Advanced VSR methods focus on harnessing temporal information through various techniques such as sliding windows [16, 24, 29, 30, 37, 57, 60, 74] and recurrent structures [5–7, 27, 28, 48, 73]. Recently, with the rapid development of Transformers in computer vision, several attempts have been made to exploit Transformers for better recovering missing details in LR sequences [31, 34, 43, 51]. However, effectively modeling and harnessing temporal relationships continues to pose an open and formidable challenge in the task of VSR.

Event cameras represent an innovative class of bio-inspired sensors capable of asynchronously detecting intensity changes in individual pixels at the microsecond level [47]. These sensors can

generate asynchronous event data, amounting to millions of events per second while maintaining robustness in HDR lighting conditions. Therefore, recent event-guided VSR methods have been proposed [19, 21, 22, 36, 67] to leverage the advantages of the events for VSR, comparing favorably to RGB-only methods. In practice, however, these event-guided VSR methods assume *strict alignment between images and events*. To substantiate this assumption, [19] leverages the CED dataset [49], which comprises aligned images and events sourced from DAVIS346. Similarly, [36] introduces the well-aligned ALPIX-VSR dataset for event-guided VSR.

In practice, however, in our daily imaging systems, especially in edge devices like dual-lens (or more) smartphones and unmanned aerial vehicles [11], deploying strictly aligned event and RGB cameras is challenging, let alone leveraging data collected from aligned cameras for downstream tasks. It is crucial to develop an asymmetric VSR algorithm based on asymmetric stereo events and RGB cameras to address this issue. However, utilizing information from different modalities and dealing with non-aligned data pose significant challenges for this task.

In this paper, to solve this new task, we propose the Asymmetric Event-guided VSR Network (AsEVSRN). AsEVSRN is the first end-to-end learning-based network that can generally be applied to super-resolve an LR video using an asymmetric event camera. Our proposed AsEVSRN introduces two key components to leverage asymmetric event streams for VSR. Firstly, our proposed content hallucination (CH) module dynamically enhances both event and RGB information by exploiting their complementary characteristics, thereby adaptively boosting representational capacity. Specifically, we adopt a dual-branch architecture to fuse event and RGB information adaptively and employ a dynamic convolution for dynamic enhancement of representational capacity. Secondly, drawing inspiration from successful practices in existing VSR methods [6], we design the event-enhanced bidirectional recurrent cells. The event-enhanced bidirectional recurrent cells align and propagate temporal features, integrating them with features extracted from content-hallucinated frames. Due to misalignment between the event stream and RGB frames, direct utilization for temporal fusion and propagation is not feasible. Therefore, our proposed event-enhanced bidirectional recurrent cells first pre-align event information with RGB views using a deformable convolution, enabling simultaneous utilization and fusion of temporal information at both the feature and pixel levels. We conduct experiments using event-RGB stereo data. Through extensive experimentation, we have quantitatively and qualitatively demonstrated the effectiveness of AsEVSRN.

In summary, our contributions can be summarized as follows:

(1) We propose AsEVSRN for super-resolving RGB frames with the guidance of the asymmetric event data. To the best of our knowledge, this is the first event-guided VSR method based on asymmetric event and RGB cameras.

(2) We propose the CH module dynamically to enhance event and RGB information by leveraging their complementary nature, thereby adaptively boosting representational capacity.

(3) We propose the event-enhanced bidirectional recurrent cells to align and propagate temporal features fused with features from content-hallucinated frames. Within these recurrent cells, event-enhanced flow facilitates simultaneous utilization and fusion of temporal information at both feature and pixel levels.

(4) Extensive experiments demonstrate that our proposed AsEVSRN is superior to the existing advanced potential methods.

## 2 Related Work

*Video Super-Resolution.* Existing RGB-only VSR methods enhance LR frames using temporal information via sliding windows [63, 65, 66, 69] and recurrent structures [6, 7]. Sliding-window techniques like 3DSRNet [24], TDAN [57], and EDVR [60] recover HR frames by predicting dynamic offsets and sampling convolution kernels from adjacent LR frames. They employ methods such as 3D convolution [24], optical flow estimation [25, 55], and deformable convolution [57, 60] to align temporal features. However, capturing long-range temporal features remains challenging for these approaches. To address this, recurrent structures-based methods [6, 13, 15, 55, 73] have been developed to model long-range temporal dependencies by utilizing hidden states to connect video frames. For example, BasicVSR [6] uses a bidirectional recurrent structure that merges forward and backward propagation features, resulting in significant performance gains. Vision Transformer-based methods [3, 34, 51, 75] have also achieved remarkable success in VSR. This paper focuses on event-guided VSR, which integrates event cameras to enhance VSR performance.

*Event-Guided VSR.* Event-guided VSR emerges as a pivotal application, leveraging the high-frame-rate motion details captured by event cameras. In event-guided VSR, consecutive frames and event data are utilized to generate HR frames. Various approaches have emerged in this domain. Jing *et al.* [19] propose a two-stage method that leverages events to interpolate the LR video, resulting in a high-frequency video that is then used to reconstruct HR frames. Kai *et al.* [22] introduce a bidirectional VSR framework, which harnesses nonlinear motion information from events to aid temporal alignment and incorporates a bidirectional cross-modal synthesis module to enhance the model's robustness to lighting variations. Lu *et al.* [36] present a joint framework that learns implicit neural representations from both RGB frames and events, enabling arbitrary-scale VSR. However, these methods presume strict alignment between the event stream and RGB images, posing practical challenges in real-world applications. For instance, in edge devices like dual-lens smartphones and unmanned aerial vehicles [11], acquiring strictly aligned event and RGB cameras can be challenging. To tackle this issue, we propose the first framework for asymmetric event-guided VSR.

*Event-Guided Video Restoration.* Event cameras have the unique capability to measure intensity changes at each pixel independently with microsecond accuracy, making them valuable for various video restoration tasks. One of the notable advantages of event cameras is their ability to provide motion information within the exposure time, which serves as a natural motion cue for deblurring [18, 50, 52, 53, 76]. In the context of video frame interpolation, the integration of event cameras has sparked innovations, such as TimeLens [59]. Subsequently, there has been a growing emphasis on designing interaction modules facilitating the exchange of information between event data and RGB frames, ultimately enhancing the performance of event-based video frame interpolation [42, 58, 68]. Additionally, event cameras have proven useful

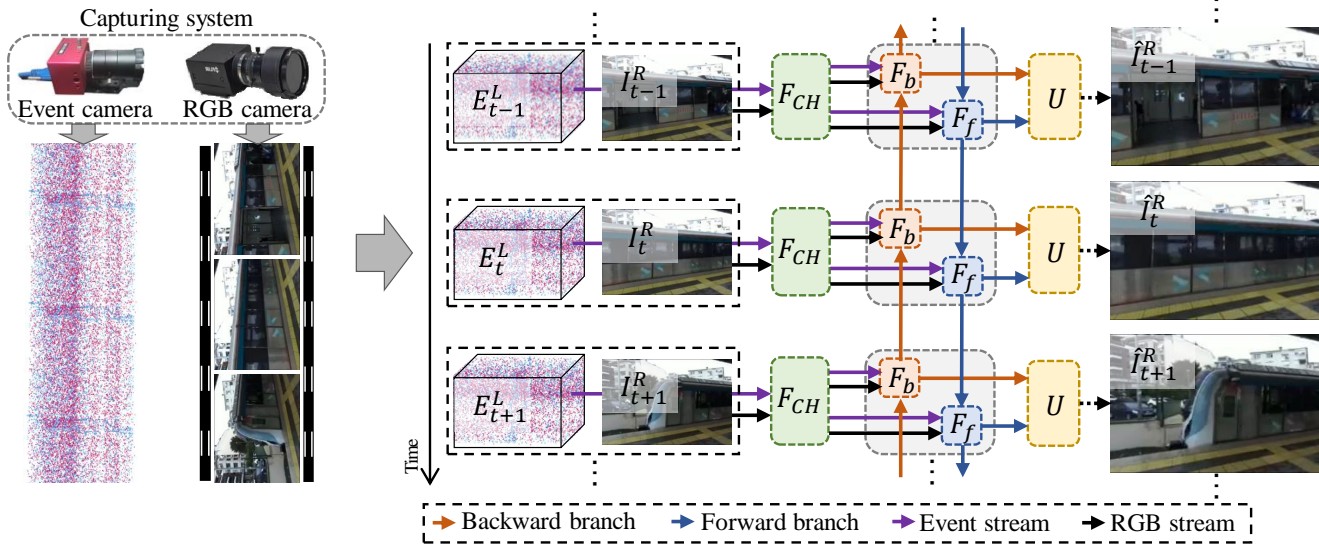

**Figure 1: Left: Illustration of the asymmetric Event and RGB cameras System. Right: Overview of the proposed AsEVSRN. The information from the left event stream and the right RGB frame stream are fed to content hallucination module, aiming to highlight valuable information while mitigating interference from misaligned data from different modalities. Then the hallucinated event and image features are fed to bidirectional recurrent cells for further alignment and aggregation. Finally, the bidirectional features are fed to the upsampling module to generate the super-resolved results. For simplicity, the upsampling residual operations are omitted in the figure.**

in correcting rolling shutter artifacts in consecutive global shutter frames [10, 35, 61, 78]. They have also been applied to tasks such as high-dynamic-range imaging [40, 46, 72, 80], deraining [53], and low-illumination enhancement [17], showcasing their versatility across various domains. In this paper, to the best of our knowledge, we propose the first event-guided VSR method based on asymmetric event and RGB cameras.

## 3 Method

### 3.1 Overview

Given a multi-modal stereo camera capturing system, where the left camera is an event camera and the right one is a normal LR RGB camera, our goal is to reconstruct consecutive HR clear right frames $\hat{I}^R = \{\dots, \hat{I}^R_{t-1}, \hat{I}^R_t, \hat{I}^R_{t+1}, \dots\}$ ($\hat{I}^R \in \mathbb{R}^{T \times sH \times sW \times 3}$) using the captured right LR frames $I^R = \{\dots, I^R_{t-1}, I^R_t, I^R_{t+1}, \dots\}$ ($I^R \in \mathbb{R}^{T \times H \times W \times 3}$) and the corresponding left event streams $\mathcal{E}^L = \{\dots, E^L_{t-1}, E^L_t, E^L_{t+1}, \dots\}$ triggered within $T$. $E^L_t$ denotes the left event stream at the time stamp $t$. $\hat{I}^R$ should be close to the ground truth $I^{R,GT} = \{\dots, I^{R,GT}_{t-1}, I^{R,GT}_t, I^{R,GT}_{t+1}, \dots\}$ ($I^{R,GT} \in \mathbb{R}^{T \times sH \times sW \times 3}$). $T$, $H$, and $W$ are the frame number, height, and width, respectively. $s$ is the upscaling factor and $T = \{\dots, t-1, t, t+1, \dots\}$. Note that, the camera system can consist of an event camera on the left and an RGB camera on the right, or vice versa with the RGB camera on the left and the event camera on the right.

Due to the fact that the event streams are not convenient for observation and processing by convolutional neural networks because of their sparse, irregular and unstructured properties, we convert event streams $\mathcal{E}^L$ into voxel grids $\mathcal{V}^L \in \mathbb{R}^{T \times H \times W \times B}$ using

temporal bilinear transformation, suitable for convolutional neural networks [36, 68]

$$\mathcal{V}(k) = \sum_i p_i \max(0, 1 - |k - \frac{t_i - t_0}{t_{N_e} - t_0}(B-1)|), \quad (1)$$

where $t_0$ and $t_{N_e}$ denote the start time and the end time of the event stream, and $N_e$ denotes the number of the event data. $t_k$ is the event firing timestamp, and $p_k$ is the polarity indicating the sign of illumination changes, respectively. The index $k$ spans from 0 to $B-1$, with $B$ set as 5 in our experiments.

Figure 1 shows an overview of the proposed AsEVSRN. AsEVSRN employs bidirectional recurrent cells $F_f$ and $F_b$ akin to the scheme proposed in [6]. However, it introduces novel elements such as extra inputs and specialized modules to harness event streams, setting it apart from prior approaches. The left event streams and the right LR frames are first converted into the feature domain using the feature encoders ($f^V_{En}$ and $f^I_{En}$), following which they are directed into the CH module, denoted as $f_{CH}(\cdot)$. This step aims to accentuate valuable information while simultaneously mitigating interference originating from misaligned data across distinct modalities. The above process can be denoted as

$$\boldsymbol{F}^E = f^V_{En}(\mathcal{V}^L), \boldsymbol{F}^I = f^I_{En}(I^R), \quad (2)$$

$$\boldsymbol{F}^{E,CH} = f_{CH}(\boldsymbol{F}^E, \boldsymbol{F}^I), \boldsymbol{F}^{I,CH} = f_{CH}(\boldsymbol{F}^I, \boldsymbol{F}^E), \quad (3)$$

Then the hallucinated event and image features ($\boldsymbol{F}^{E,CH}$ and $\boldsymbol{F}^{I,CH}$) are fed to recurrent cells, and for a time step $t$, each recurrent cell $F_f$ or $F_b$ not only takes the hallucinated event and image features at the current time step, but also the corresponding features ($h^f_{t-1}$ and $h^b_{t+1}$) propagated from its neighbors. Moreover, each recurrent

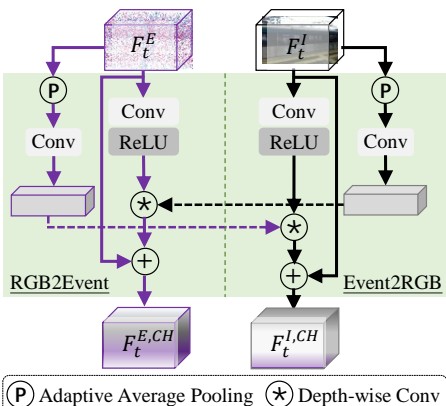

**Figure 2: The structure of the content hallucination module.**

cell propagates the resulting features $h_t^{\{f,b\}}$ to the next cell. The above process can be denoted as

$$h_t^f = F_f(F_t^{E,CH}, F_t^{I,CH}, h_{t-1}^f),$$
$$h_t^b = F_b(F_t^{E,CH}, F_t^{I,CH}, h_{t+1}^b). \tag{4}$$

To generate a super-resolved output $\hat{I}_t^R$ at timestamp $t$, the up-sampling module $U$ incorporates multiple convolutional layers along with pixel-shuffle operations. This module takes intermediate features $h_t^{f,b}$ and the LR frame $I_t^R$ as inputs, yielding the final super-resolved frame $\hat{I}_t^R$

$$\hat{I}_t^R = U([h_t^f, h_t^b, I_t^R]) + (I_t^R) \uparrow_s, \tag{5}$$

where $[\cdot, \cdot]$ is the concatenation operation, and $(\cdot) \uparrow_s$ is the bilinear upsampling operation with a scaling factor of $s$.

## 3.2 Content Hallucination

In scenarios involving LR scenes, both event streams and LR images inherently contain noise in the form of missing details and artifacts, respectively. To tackle this challenge and leverage the complementary nature of information across modalities, we propose the CH module (see Figure 2). Specifically, the CH module adopts a dual-branch structure (*i.e.*, RGB2Event, and Event2RGB branches), enabling the simultaneous hallucination of representations for two modal features.

Given the left event feature $F_t^E$ and the right frame feature $F_t^I$, the proposed CH module initially employs an adaptive estimation process to derive dynamic filters of high-level contextual information independently for each modality branch. Subsequently, these dynamic filters are utilized to enhance the features of the corresponding modality, facilitating cross-modal feature refinement and integration for improved representation learning. We have

$$K_t^E = \phi_3(P(F_t^E)), K_t^I = \phi_3(P(F_t^I)),$$
$$F_t^{E,CH} = K_t^I \circledast A(\phi_3(F_t^E)) + F_t^E, \tag{6}$$
$$F_t^{I,CH} = K_t^E \circledast A(\phi_3(F_t^I)) + F_t^I,$$

where $\phi_3(\cdot)$ denotes the convolution operation, $P(\cdot)$ denotes the adaptive average pooling operation, $A(\cdot)$ denotes the ReLU activation operation, and $\circledast$ is the depth-wise convolution. Using a

learned dynamic filter from one modality to modulate the feature representation of another, the proposed CH module enhances valuable information while mitigating interference. This module facilitates the refinement and integration of features across modalities, thereby promoting more effective representation learning in the task of event-guided VSR.

## 3.3 Bidirectional Recurrent Cells

In VSR, exploiting temporal information is crucial, particularly in asymmetric event-guided VSR scenarios. The bidirectional propagation scheme has been widely acknowledged for its effectiveness [6, 7, 54]; therefore, we adopt it in AsEVSRN. Specifically, we draw inspiration from the bidirectional recurrent cells employed in BasicVSR to implement the utility of bidirectional propagation.

In each recurrent cell $F_f(\cdot)$ and $F_b(\cdot)$, a flow estimation network (*e.g.*, SpyNet[44]) is typically employed to estimate the optical flow between the LR frame $I_t^R$ at the current time step and $I_{t\pm1}^R$ at the previous or the next time steps for alignment and further processing. The optical flow estimation can be denoted as

$$O_t^f = \text{SpyNet}(I_t^R, I_{t-1}^R), O_t^b = \text{SpyNet}(I_t^R, I_{t+1}^R). \tag{7}$$

However, under the practical setting of asymmetric event-guided VSR, the conventional approach described above fails to fully exploit the asymmetric information from events. Considering that the event information can effectively assist in aligning and fusing multiple frames [9, 18, 33, 42, 59], we propose the event-enhanced bidirectional recurrent cells.

Although the event stream contains crucial temporal information and cues for alignment, asymmetric events may adversely affect the alignment of RGB frames [4, 26, 27]. Therefore, we first employ deformable convolution to align the event stream with the RGB frames in feature space. Specifically, taking the forward recurrent cell as an example (see Figure 3), the content-hallucinated features $F_t^{E,CH}$ and $F_t^{I,CH}$ are first concatenated, followed by a convolutional layer to generate the RGB-aware offset map $\Delta P_t^f$. $\Delta P_t^f$ and $F_t^{E,CH}$ are then fed to the deformable convolution layer, resulting in $F_t^{E,R}$. This procedure can be denoted as

$$\Delta P_t^f = \text{Conv}([F_t^{I,CH}, F_t^{E,CH}]), \tag{8}$$
$$F_t^{E,R} = \text{DConv}(F_t^{E,CH}, \Delta P_t^f), \tag{9}$$

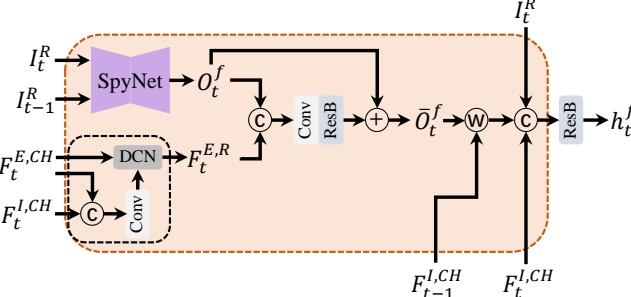

**Figure 3: The structure of the event-enhanced forward recurrent cell. The event-enhanced backward recurrent cell one can be obtained in a similar way.**

**Table 1: Quantitative comparison on the KITTI 2012 and KITTI 2015 datasets for $4\times$ asymmetric event-guided VSR in terms of PSNR and SSIM. The best results are marked in bold, the second ones are marked with underlines, and the third ones are marked with wavy lines. The number of parameters (M) and runtime (ms) are calculated using an NVIDIA GTX 1080Ti GPU for $4\times$ asymmetric event-guided VSR (spatial resolution: $48 \times 48 \rightarrow 192 \times 192$).**

| | Method | #Params (M) | Runtime (ms) | KITTI 2012 | | KITTI 2015 | |
|---|---|---|---|---|---|---|---|
| | | | | PSNR | SSIM | PSNR | SSIM |
| SISR | Bicubic | - | - | 25.36 | 0.7530 | 25.76 | 0.7613 |
| | SwinIR | 11.504 | 240.99 | 29.14 | 0.8618 | 29.98 | 0.8700 |
| | SRFormer | 10.396 | 291.91 | 29.19 | 0.8623 | 30.00 | 0.8697 |
| VSR | VSRNet | 0.439 | 37.63 | 27.65 | 0.8253 | 28.18 | 0.8308 |
| | DUF | 5.822 | 78.42 | 29.05 | 0.8585 | 29.75 | 0.8642 |
| | TOF | 1.406 | 104.10 | 28.74 | 0.8512 | 28.07 | 0.8286 |
| | EDVR | 20.699 | 71.69 | 30.27 | 0.8888 | 30.80 | 0.8869 |
| | BasicVSR | 6.291 | 21.31 | 30.69 | 0.8970 | 31.17 | 0.8938 |
| | BasicVSR++ | 7.323 | 20.99 | 30.83 | 0.8992 | 31.30 | 0.8961 |
| | TTVSR | 3.450 | 19.32 | 30.74 | 0.8983 | 31.11 | 0.8935 |
| | PSRT | 13.367 | 187.09 | 30.48 | 0.8955 | 31.02 | 0.8936 |
| | IART | 13.411 | 193.97 | 30.52 | 0.8980 | 31.01 | 0.8951 |
| Event-guided VSR | EGVSR | 2.574 | 44.91 | 30.41 | 0.8946 | 30.90 | 0.8971 |
| | EBVSR | 12.151 | 46.76 | 30.89 | 0.8997 | 31.31 | 0.8965 |
| | **AsEVSRN (Ours)** | 9.648 | 49.37 | **31.17** | **0.9052** | **31.91** | **0.9048** |

where $[\cdot, \cdot]$, $\text{Conv}(\cdot)$ and $\text{DConv}(\cdot)$ denotes the concatenation operation, the convolution layer and the deformable convolution layer, respectively. The aligned event feature $F_t^{E,R}$ is then employed for flow refinement. On one hand, it is used to mitigate the influence of low-resolution frames on flow estimation. On the other hand, the motion information from events is utilized to further optimize the optical flow $O_t^f$. To obtain refined optical flow, we directly concatenate $F_t^{E,R}$ and $O_t^f$ and feed them into a convolutional layer and a residual block. We then utilize residual connections to obtain $\overline{O}_t^f$

$$\overline{O}_t^f = \text{ResB}(\text{Conv}([F_t^{E,R}, O_t^f])) + O_t^f, \quad (10)$$

where $ResB(\cdot)$ is the residual block.

To obtain the temporally aggregated feature $h_t^f$, we utilize the refined optical flow to warp the RGB feature through a warping operation. Then, we concatenate the warped result with $I_t^R$ and feed it into a residual block. This design effectively leverages information from both the feature domain and the pixel domain [68], enhancing the recurrent cell's representation capability. Formally, we have

$$h_t^f = \text{ResB}([I_t^R, F_t^{I,CH}, \text{Warp}(F_{t-1}^{I,CH}, \overline{O}_t^f)]), \quad (11)$$

where $\text{Warp}(\cdot, \cdot)$ denotes the warping operation.

The event-enhanced backward recurrent cell can be obtained using a similar method to obtain $h_t^b$.

## 4 Experiments

### 4.1 Experimental Settings

*Datasets.* We train and evaluate our proposed AsEVSRN on KITTI 2012 [2] and KITTI 2015 [39] datasets. KITTI 2012 is a real-world dataset with street views from a driving car. It consists of 194 training stereo video clips and 195 testing clips, each with a

resolution of $1242 \times 375$ pixels and a total of 21 frames per clip. KITTI 2015 is also a real-world dataset that shares the same shooting conditions as KITTI 2012 but with higher quality. It contains 200 training stereo video clips and 200 testing stereo video clips. The resolution and frame number are the same as KITTI 2012. Without loss of generality, we transform the left view of KITTI 2012 and KITTI 2015 into event data, while keeping the right view unchanged. This creates asymmetric event-RGB inputs. We first utilize the pre-trained RIFE interpolation model [14] to generate additional left-view frames at a $4\times$ higher frame rate. Then we use the event camera simulator ESIM [45], to simulate events from the interpolated high-frame-rate left videos.

*Training Settings.* During the training stage, we follow the division of the training sets of the KITTI 2012 and KITTI 2015 datasets. We utilize bicubic downsampling by a factor of 4 on the left and right view video frames to obtain LR frames. In other words, we set $s = 4$. The proposed AsEVSRN aims to learn the mapping relationship from low-resolution frames to high-resolution frames. Given the ground-truth frame $\mathcal{I}^{R,GT}$ and the super-resolved results $\hat{\mathcal{I}}^R$ generated by our proposed AsEVSRN, we adopt the simple but effective Charbonnier loss [60] to train it from scratch, which can be described as:

$$\mathcal{L} = \sqrt{\left\| \mathcal{I}^{R,GT} - \hat{\mathcal{I}}^R \right\|^2 + \varepsilon^2}, \quad (12)$$

where $\varepsilon$ is set to $1e - 6$ in our experiments. Following previous works, we use a pre-trained SpyNet to estimate optical flow in the event-enhanced bidirectional recurrent cells. We utilize the Adam optimizer with parameters $\beta_1 = 0.9$ and $\beta_2 = 0.999$, and we utilize the Cosine Annealing scheduler for optimization. Each mini-batch consists of 6 samples. The input patch size is set to $64 \times 64$. Experiments are conducted using PyTorch [85] on two NVIDIA

3090 GPUs. We fix the weights of the pre-trained SpyNet in the first 5K iterations, and the total number of iterations is 300K.

*Inference Settings.* During the testing stage, we follow the division of the training sets of the KITTI 2012 and KITTI 2015 datasets. To quantitatively evaluate the reconstructed HR videos, we choose Peak Signal-to-Noise Ratio (PSNR) and Structural Similarity Index (SSIM) on Y channel as metrics. The temporal consistency can be analyzed by evaluating the estimated optical flow of the reconstructed HR videos and the extracted temporal profiles.

## 4.2 Quantitative and Qualitative Comparisons

We compare the proposed AsEVSRN with a wide range of potential methods that could be used to address asymmetric event-guided VSR, aiming to explore as many diverse and rich approaches as possible. (1) Single image SR (SISR) methods: Bicubic, SwinIR [32], and SRFormer [79]. Specifically, we process each LR frame sequentially through the SISR network for reconstruction, and then assemble the reconstructed frames to form an HR video. (2) VSR methods: VSRNet [23], DUF [20], TOF [71], EDVR [60], BasicVSR [6], BasicVSR++ [7], PSRT [51], TTVSR [34], and IART [70]. In particular, we exclude the event stream and solely feed the LR video frames into the VSR network for reconstruction, resulting in the final reconstructed video output. (3) Event-guided VSR methods: as this direction is relatively less explored, we compare our AsEVSRN with EGVSR and EBVSR to provide a thorough evaluation. It is important to note that for fair comparison, we retrain all these methods on the KITTI 2012 and KITTI 2015 datasets using their publicly released codes. We refrain from using fine-tuning or directly utilizing pre-trained models on Vimeo90K [71] or REDS [41].

*Quantitative Results.* Table 1 shows the quantitative comparisons on various testsets in terms of PSNR and SSIM. From the table, we can draw several conclusions. Firstly, VSR methods generally outperform SISR methods in terms of PSNR and SSIM. This indicates that the temporal information provided by video sequences helps in achieving better reconstruction quality compared to SISR methods. For instance, BasicVSR and its variants consistently outperform SwinIR, which is a representative leading SISR method. Secondly, event-guided VSR shows potential for achieving higher performance compared to traditional VSR methods. For instance, methods like EBVSR, which incorporates event information, exhibit higher PSNR and SSIM values than some traditional VSR methods like BasicVSR++ and TTVSR. For example, for EBVSR compared to TTVSR on the KITTI 2012 dataset, there is a PSNR increase of 0.15 dB and a SSIM increase of 0.0014. On the KITTI 2015 dataset, EBVSR shows a PSNR increase of 0.20 dB and an SSIM increase of 0.0030 compared to TTVSR. Lastly, our proposed AsEVSRN demonstrates superior performance compared to both traditional VSR methods and other event-guided VSR methods. Specifically, AsEVSRN achieves the highest PSNR and SSIM values among all methods evaluated on both KITTI 2012 and KITTI 2015 datasets. For instance, AsEVSRN achieves a PSNR of 31.17 dB on KITTI 2012, outperforming all other baseline methods.

*Computational Efficiency.* We compare AsEVSRN to other methods in terms of the number of parameters and the runtime. Results

are listed in Table 1. Comparing the number of parameters and runtime among different methods, we observe that AsEVSRN achieves competitive performance with fewer parameters and comparable runtime. Specifically, AsEVSRN has 9.648M parameters and a runtime of 49.37 ms, while EBVSR, which has a similar performance, requires 12.151M parameters and a runtime of 46.76 ms. Despite having fewer parameters, AsEVSRN achieves higher PSNR and SSIM scores compared to EBVSR. This indicates that AsEVSRN effectively utilizes parameter efficiency to improve reconstruction quality, demonstrating its superiority in terms of performance-complexity trade-off. Therefore, even under similar computational constraints, AsEVSRN outperforms other methods in terms of PSNR and SSIM on both KITTI 2012 and KITTI 2015 datasets.

*Qualitative Results.* In Figure 4, we present visual comparisons between the results obtained by our proposed AsEVSRN and those of other competing baselines on the KITTI 2012 dataset. It is evident from these visual comparisons that our proposed AsEVSRN method outperforms the baseline methods, yielding superior qualitative results characterized by more accurate details and significantly reduced blurring artifacts. For instance, in the restoration of fine texture details, our AsEVSRN excels in reconstructing the intricate brick patterns on the rooftops, while artifacts are visibly present in the results produced by other methods. Furthermore, AsEVSRN demonstrates a clearer restoration of textural details compared to other methods, which often generate blurry results. This is particularly noticeable in regions with complex textures and high-frequency details, where our method successfully recovers fine structures without introducing unwanted noise or smoothing effects. These qualitative improvements underscore the efficacy of our proposed architecture in enhancing the visual quality of super-resolved video content.

*Temporal Consistency.* To evaluate the temporal consistency of the super-resolved video clips, we estimate the optical flow on the KITTI 2012 dataset using the advanced RAFT algorithm [56]. Specifically, we compute the optical flow between consecutive frames to assess the motion coherence of the reconstructed sequences. As illustrated in Figure 5, the optical flow estimated from the results produced by our proposed AsEVSRN is remarkably close to that obtained from the original HR frames. This observation substantiates the superior temporal consistency and motion accuracy of our method, demonstrating its effectiveness in generating high-quality, temporally coherent super-resolved video content.

## 4.3 Ablation Studies

To analyze the effectiveness of the proposed AsEVSRN, we conduct the following experiments on KITTI 2012.

*Effectiveness of Two Modules in AsEVSRN.* The CH module and the event-enhanced bidirectional recurrent cells are two core modules in AsEVSRN. We analyze the performance of each component by removing different modules and replacing them with residual blocks of equivalent parameter amount. Table 2 presents the effectiveness of the CH module and the event-enhanced bidirectional recurrent cells in terms of PSNR and SSIM. Each method is evaluated with different combinations of these modules. Firstly, when excluding the CH module (AsEVSRN-w/o-$F_{CH}$), the PSNR is 30.72 dB and SSIM

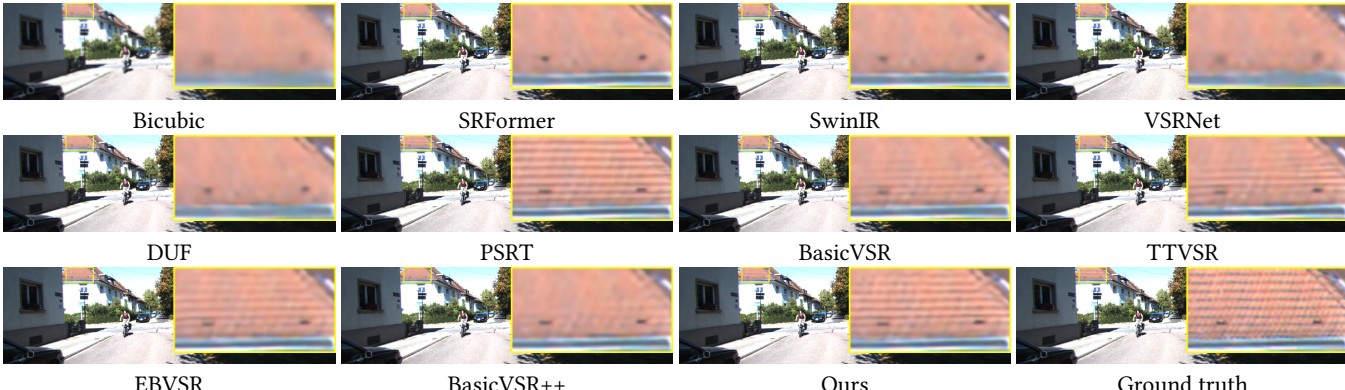

**Figure 4: Visual comparison on the** $4\times$ **asymmetric event-guided VSR task. Frames are from the** KITTI 2012 **dataset.**

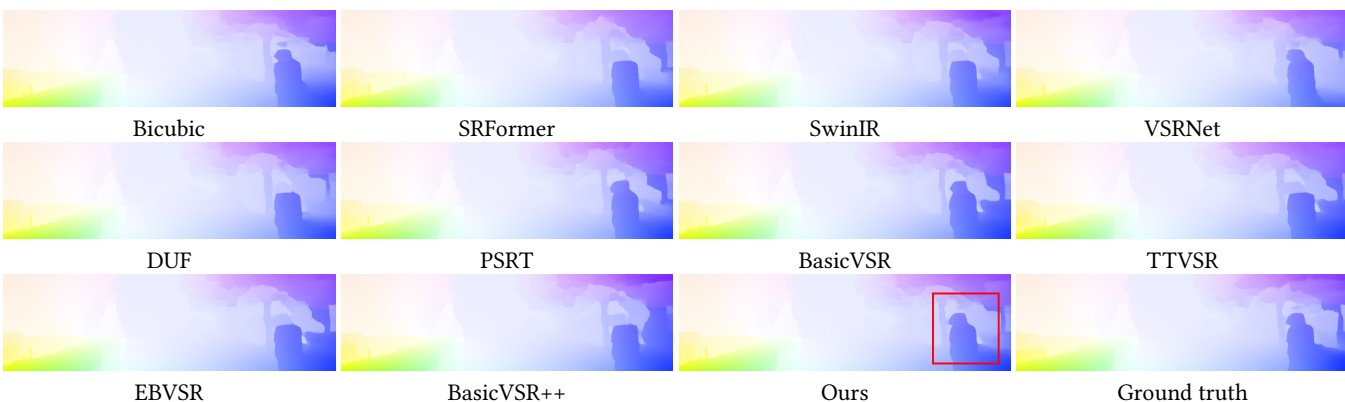

**Figure 5: Temporal consistency comparison on the** $4\times$ **asymmetric event-guided VSR task. We show the estimated optical flow of the results from different methods using the pre-trained RAFT [56].**

is 0.8975. This indicates that the absence of the CH module leads to a decrease in performance compared to the full AsEVSRN method. Secondly, omitting the backward recurrent cell ($F_b$) while including the CH module (AsEVSRN-w/o-$F_b$) results in a slight improvement in PSNR to 30.95 dB and SSIM to 0.9011 compared to the case without the CH module. Similarly, excluding the forward recurrent cell ($F_f$) while keeping the CH module (AsEVSRN-w/o-$F_f$) yields a PSNR of 30.96 dB and SSIM of 0.9014, slightly higher than the previous case. Finally, the full AsEVSRN method, incorporating both the CH module and bidirectional recurrent cells, achieves the highest PSNR of 31.17 dB and SSIM of 0.9052, demonstrating the effectiveness of both modules in enhancing the performance of the proposed method.

*Investigation of the CH Module.* The CH module aims at leveraging the complementary nature of information across the event and RGB modalities while enhancing the representation ability. To showcase its effectiveness, we design and analyze several variants: (1) CH-w/o-dynamicfilter: we replace the dynamic filter with a simple addition operation. (2) CH-w/o-RGB2Event: in this variant, we directly remove the RGB2Event branch. (3) CH-w/o-Event2RGB: this variant involves the direct removal of the Event2RGB branch.

**Table 2: Effectiveness of the CH module and the event-enhanced bidirectional recurrent cells.**

| Method | $F_{CH}$ | $F_{fb}$ | | PSNR | SSIM |
| --- | --- | --- | --- | --- | --- |
| | | $F_b$ | $F_f$ | | |
| AsEVSRN-w/o-$F_{CH}$ | ✗ | ✓ | ✓ | 30.72 | 0.8975 |
| AsEVSRN-w/o-$F_b$ | ✓ | ✗ | ✓ | 30.95 | 0.9011 |
| AsEVSRN-w/o-$F_f$ | ✓ | ✓ | ✗ | 30.96 | 0.9014 |
| AsEVSRN | ✓ | ✓ | ✓ | 31.17 | 0.9052 |

**Table 3: Effectiveness of the designs in the CH module.**

| Method | PSNR | SSIM |
| --- | --- | --- |
| CH-w/o-dynamicfilter | 30.93 | 0.8995 |
| CH-w/o-RGB2Event | 30.99 | 0.9016 |
| CH-w/o-Event2RGB | 31.00 | 0.9020 |
| CH Module | 31.17 | 0.9052 |

Results are shown in Table 3. The results show that including both

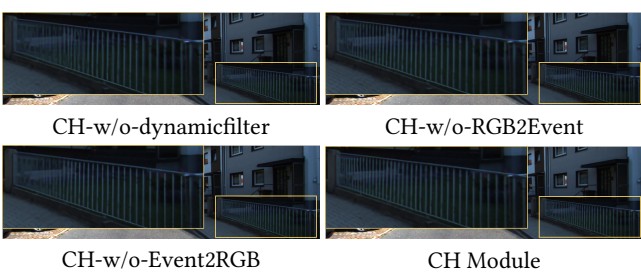

CH-w/o-dynamicfilter    CH-w/o-RGB2Event

CH-w/o-Event2RGB    CH Module

**Figure 6: Visual comparison on removing different parts in the CH module. Please zoom in for better viewing.**

**Table 4: Effectiveness of the designs in the event-enhanced bidirectional recurrent cells.**

| Method | PSNR | SSIM |
|---|---|---|
| Cell-w/o-DCN | 30.96 | 0.9013 |
| Cell-w/o-$F_t^{E,R}$ | 31.00 | 0.9021 |
| Cell-w/o-$\overline{O}_t^f$ | 31.04 | 0.9033 |
| Cell-w/o-$F_t^{I,CH}$ | 31.14 | 0.9036 |
| Cell-w/o-$I_t^R$ | 31.12 | 0.9035 |
| Cell | 31.17 | 0.9052 |

**Table 5: Embedding the components of the AsEVSRN into existing baseline methods, *i.e.*, BasicVSR and EBVSR. † denotes the method with the CH module and event-enhanced bidirectional recurrent cells.**

| Method | PSNR | SSIM |
|---|---|---|
| BasicVSR | 30.69 | 0.8970 |
| BasicVSR† | 30.92 | 0.9001 |
| EBVSR | 30.89 | 0.8997 |
| EBVSR† | 31.03 | 0.9029 |

RGB-to-Event and Event-to-RGB fusion, along with dynamic filtering in the CH module enhances performance, as evidenced by the highest PSNR and SSIM values compared to the configurations with individual components excluded. In Figure 6, we also visualize the results after removing different parts in the CH module. The removal of dynamic convolution resulted in an overall deterioration in performance. Furthermore, eliminating the RGB2event branch lead to a degradation in the reconstructed details, while removing the event2RGB branch caused the results to become blurrier. These observations align with the findings presented in Table 3.

*Investigation of the Event-Enhanced Bidirectional Recurrent Cells.* The event-enhanced bidirectional recurrent cells aim at leveraging the event information for RGB feature fusion and propagation. To showcase its effectiveness, we design and analyze several variants: (1) Cell-w/o-DCN: in this variant, we directly remove the $G^2DT$ module. (2) Cell-w/o-$F_t^{E,R}$: we feed the content-hallucinated event feature to the following parts directly. (3) Cell-w/o-$\overline{O}_t^f$: we utilize the optical flow estimated by SpyNet directly. (4) Cell-w/o-$F_t^{I,CH}$:

we perform the warping operation at the pixel level. (5) Cell-w/o-$I_t^R$: we perform the warping operation at the feature level. Table 4 presents the effectiveness of different designs in the event-enhanced bidirectional recurrent cells based on PSNR and SSIM metrics. The results show that each design variation contributes to improving performance, with the complete cell achieving the highest PSNR of 31.17 dB and SSIM of 0.9052 compared to its variants without specific components.

*Embedding the Components of the AsEVSRN into Existing Baseline Methods.* Table 5 presents the results after integrating two important components into BasicVSR and EBVSR. We observe that incorporating these components leads to improvements in both PSNR and SSIM metrics for both methods. Specifically, BasicVSR with the integrated components achieves a PSNR of 30.92 dB and SSIM of 0.9001, while EBVSR with the integrated components achieves a PSNR of 31.03 dB and SSIM of 0.9029. These results indicate that the integration of the components enhances the performance of both BasicVSR and EBVSR methods, demonstrating the effectiveness of the proposed components.

## 4.4 Limitation

While our AsEVSRN demonstrates promising results, there are still challenges that need to be addressed. For instance, we encounter difficulties in reconstructing small objects, as elaborated in the supplementary material. These small objects often require finer detail recovery, which can be challenging due to the limited amount of information available at lower resolutions.

## 5 Conclusion

In this paper, we address the challenge of performing asymmetric event-guided VSR for the first time, introducing AsEVSRN tailored specifically for this novel task. The AsEVSRN leverages two specialized designs: a content hallucination module that dynamically enhances event and RGB information, boosting representational capacity, and event-enhanced bidirectional recurrent cells that align and propagate temporal features fused with content-hallucinated frames. These cells employ event-enhanced flow for the simultaneous utilization and fusion of temporal information at both the feature and pixel levels. The AsEVSRN consistently generates superior results both quantitatively and qualitatively.

## Acknowledgments

We acknowledge funding from National Natural Science Foundation of China under Grants 62131003, 62021001 and 61901435.

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
