# OpenReview forum: "Asymmetric Event-Guided Video Super-Resolution"
_acmmm.org/ACMMM/2024/Conference — MM2024 Poster_

### Official Review · Reviewer_Db9H · 2024-05-16

**Rating:** 2
**Confidence:** 3

**Summary:**

This work introduces a new video super-resolution (VSR) method leveraging the capabilities of asymmetric event cameras paired with RGB cameras. This new approach, named Asymmetric Event-guided VSR Network (AsEVSRN), does not require strict alignment between the event and RGB cameras, which broadens the applicability of VSR technology in practical scenarios like smartphones and drones. The technique utilizes a content hallucination module to dynamically enhance the information from both event and RGB data and employs event-enhanced bidirectional recurrent cells to manage temporal features effectively. This approach allows for superior handling of video data with high dynamic range and motion, delivering enhanced resolution and detail.

**Strengths:**

- By integrating event-enhanced bidirectional recurrent cells, the method improves the temporal consistency of the super-resolved video.
- The proposed AsEVSRN demonstrates superior quantitative and qualitative performance over existing VSR methods.

**Limitations:**

- Figure 1 shows the illustration of the asymmetric event and RGB camera system. Maybe the definition of asymmetric is a little bit ambiguous. I note that the feature fusion in Figure 2 is adopted with the features of the event and RGB at the same time, e.g. t or t+1. How to determine the concept of asymmetric.

- As for the method section, the content hallucination module fuses the features of the event and RGB through a depthwise convolution. However, how do you perform depthwise convolution between two features rather than features and kernels in Eq (6), where K and F seem to be features?

- The design of the entire framework is less explainable. It is more module accumulation and lacks a core motivation to design the framework like Figure 1.

- In Table 1, the proposed method, AsEVSRN, outperforms EBVSR on the two datasets with fewer parameters (9.648 vs. 12.151). However, it costs more running time (49.37). Which part of the entire framework is more time-consuming? And whether this part mainly improves the PSNR or SSIM. More details need to be explained.

- For the qualitative evolution, e.g. Figure 4 and Figure 5, it is not obvious that the proposed method achieves better results compared to other methods in terms of generation quality. Take a picture as an example, ‘ours’ looks not better than SwinIR in Figure 5. Therefore, I am also suspicious of the results for video super resolution.

**Suitability:**

2

---

### Official Review · Reviewer_a5Kf · 2024-05-24

**Rating:** 3
**Confidence:** 3

**Summary:**

The paper introduces the new task of Asymmetric Event-Guided Video Super-Resolution, focusing on leveraging asynchronous event cameras for video super-resolution (VSR) tasks. The main motivation is to address the limitations of existing event-guided VSR methods that assume strict alignment between event and RGB cameras, which is often not feasible in high-resolution devices like dual-lens smartphones and unmanned aerial vehicles. To tackle this challenge, the authors propose an Asymmetric Event-guided VSR Network (AsEVSRN) that incorporates two key components: a content hallucination module to enhance event and RGB information and event-enhanced bidirectional recurrent cells for aligning and propagating temporal features. Experimental results demonstrate the superiority of AsEVSRN in producing high-quality results. The key contributions of the paper include the proposal of AsEVSRN for asymmetric event-guided VSR, the introduction of the content hallucination module, and the design of event-enhanced bidirectional recurrent cells for temporal fusion.

**Strengths:**

- Introduce a new task: Asymmetric Event-Guided Video Super-Resolution.
- Propose AsEVSRN, including two novel components.
- Demonstrate better performance of AsEVSRN compared to traditional VSR methods and other event-guided VSR methods,
- Showcase good performance-complexity trade-off.
- Effectively integrate components into existing baseline methods, leading to improvements in both PSNR and SSIM metrics.

**Limitations:**

## No video results.
Although some frames are shown in the paper to demonstrate temporal consistency, it is hard to get a sense of from a single frame. Instead, we perceive temporal consistency naturally from videos. However, I did not see any video in the supp. A suggestion for rebuttal is embedding a video in the Acrobat PDF, without using web links.
## Not convincing/unfair quantitative results.
In Table 1, BasicVSR, BasicVSR++ and TTVSR have fewer parameters and less than a half running time. It is unfair for such a comparison. What about aligning the runtime to 49.37 for BasicVSR, BasicVSR++ and TTVSR to see how they perform? Similar for EGVSR and EBVSR.
## Motivation justification
How challenging to align asynchronous event stream with video RGB frames? I did not see any illustration on this. If it is simple, this task seems to be unnecessary.
## Request to add a baseline
Following the motivation justification, it seems that one can just use accumulated events in an adjacent frame time window (e.g. LNES) as a simple alignment to RGB frames. How effective is this baseline?

**Suitability:**

3

---

### Official Review · Reviewer_HjqY · 2024-05-25

**Rating:** 5
**Confidence:** 3

**Summary:**

The paper introduces an innovative video super-resolution technique that eliminates the need to align color and event streams beforehand. In other words, no prior calibration is necessary to transform event frames to align with color frames. The authors adopt a strategy of individually hallucinating both streams, leveraging on their complementary characteristics. They also put forward the use of bidirectional recurrent cells to align and propagate temporal features, integrating them with features derived from content-hallucinated frames.

**Strengths:**

Compared to existing work, the proposed method requires no prior calibration of the event camera and RGB camera for the video super-resolution task. I believe this is a key strength of the work, enhancing its practicality for real-world applications.

The paper makes solid technical contributions with the AsEVSRN model, which uses a Content Hallucination (CH) module to dynamically enhance both event and RGB information.

The approach significantly outperforms all existing methods in Single Image Super-Resolution (SISR), Video Super-Resolution (VSR), and Event-guided VSR, marking a significant contribution. Additionally, the paper includes strong evaluations, both qualitative and in terms of ablation studies, to establish the effectiveness of the approach.

**Limitations:**

This approach may lack generalizability, as the authors have not demonstrated how their model performs when the alignment between the event camera and color camera is changed from the training set. As an augmentation method, random transformations could be applied to alter the alignment between color and event streams to evaluate robustness.

The authors mentioned they used 3090 GPUs for their training and inference; however, Table 1 states that the latency reported is measured on a 1080 Ti GPU. The authors need to clarify which GPU was used to measure the latency of which model.

Authors may consider moving equation 13 to the Method section, as it represents the loss function used for training the VSR model.

The discussion of limitations was moved to supplementary material, but including a concise version in the main paper would aid readers in better understanding the method.

The authors have not reported the latency for the preprocessing task, which converts the event streams to voxel grids. This omission may give readers a clouded understanding of the real-time performance of the system. Additionally, it is important to verify whether the run-time of the compared baselines, specifically Event-guided VSR, includes the latency for preprocessing to the required event representation.

**Suitability:**

3

---

### Meta-Review · Area_Chair_dzaj · 2024-07-01

**Recommendation:** Accept (Poster)
**Confidence:** 4

**Metareview:**

This paper got mixed reviews. Although two reviewers had negative comments on the paper, most of their comments were about some missing baselines and some details about the paper, which had been partially addressed in the authors' rebuttal. By taking all reviewers' comments into consideration, I would like to recommend acceptance of the paper.